# A Post-Authorisation Safety Study of a Respiratory Syncytial Virus Vaccine in Pregnant Women and Their Offspring in a Real-World Setting: Generic Protocol for a Target Trial Emulation

**DOI:** 10.3390/vaccines13030272

**Published:** 2025-03-05

**Authors:** Odette de Bruin, Linda Nab, Jungyeon Choi, Oisin Ryan, Hae-Won Uh, Fariba Ahmadizar, Shahar Shmuel, Heather Rubino, Kitty Bloemenkamp, Cynthia de Luise, Miriam Sturkenboom

**Affiliations:** 1Department of Data Science & Biostatistics, Julius Global Health, University Medical Center Utrecht (UMCU), 3584CG Utrecht, The Netherlands; 2Department of Obstetrics, Division Woman and Baby, Wilhelmina Children’s Hospital, University Medical Center Utrecht (UMCU), 3584CG Utrecht, The Netherlands; 3Safety Surveillance Research Worldwide Medical and Safety, Pfizer, Inc., New York, NY 10001-2192, USA

**Keywords:** RSV, target trial emulation, safety, maternal vaccination, pregnancy, protocol

## Abstract

**Background:** Assessing the real-world safety of preventive products against respiratory syncytial virus (RSV) in pregnant women holds significant public health implications, especially as vaccination programs become more widespread. This generic protocol describes a post-authorisation safety study (PASS) to evaluate the safety of RSV vaccination in pregnant women using a target trial emulation framework. **Methods:** This generic protocol, adapted from an ongoing PASS, is designed using the target trial emulation framework to evaluate the safety of an RSV vaccine in pregnant women. Emulating target trial conditions have the ability to minimise confounding and bias. In this pragmatic real-world observational study, RSV-vaccinated pregnant women are matched (1:N) with unexposed women based on gestational age, calendar time, maternal age, immunocompromised status, and high-risk pregnancy. Key adverse outcomes include preterm birth, stillbirth, hypertensive disorders of pregnancy, Guillain-Barré Syndrome (GBS), low birth weight (LBW), and small for gestational age (SGA). Future studies may add additional outcomes per vaccine risk profile and Global Alignment of Immunization safety Assessment (GAIA) recommendations. Distinguishing outcomes measured during pregnancy from those assessed at or after birth is crucial for analysis and interpretation. **Conclusions:** This protocol offers a structured approach to evaluating the safety of RSV vaccines in pregnant women. It aims to guide researchers in designing studies and should be adapted to specific settings and data availability.

## 1. Introduction

Respiratory syncytial virus (RSV) is a major cause of lower respiratory tract infections (LRTIs) in young children. In 2019 alone, RSV was responsible for 33 million LRTIs globally, leading to 3.6 million hospitalisations and around 100,000 deaths [1]. Notably, about half of the global burden of RSV falls upon infants under the age of 6 months, with a significant number occurring during the neonatal period [1]. Given the substantial impact of RSV, introducing an RSV vaccine for maternal immunisation during pregnancy may offer promise in reducing the burden of RSV-associated illness in infants [2]. The mechanism underlying maternal immunisation involves transferring antibodies across the placenta to protect the infant during its vulnerable neonatal period and beyond [3].

The development of RSV vaccines is progressing swiftly, with multiple candidate vaccines advancing through phases 2 and 3 of development [4]. In 2023, the first maternal RSV vaccine was approved by the European Medicines Agency (EMA), the Food and Drug Administration (FDA) and the Medicines and Healthcare products Regulatory Agency (MHRA) [5,6,7]. These approvals in pregnant women in the EU, US and UK are supported by evidence from a global randomized, double-blinded, and placebo-controlled trial (MATISSE). This trial assessed the vaccine’s efficacy, safety, and immunogenicity against RSV-related medically attended LRTI in infants born to vaccinated healthy women during weeks 24 to 36 of gestation. The study enrolled 7358 pregnant women, receiving either the RSV vaccine or placebo. At the time of analysis, 5654 infants (79%) completed 6-month follow-up after birth, with follow-up ongoing for a total of 24 months [8].

Immunocompromised pregnant women and those with high-risk pregnancies were excluded from the trial [8]. As a result, there is a need for further post-authorisation safety studies (PASS) to understand the vaccine’s safety profile in these populations. Additionally, Guillain-Barré Syndrome (GBS), a rare neurological disorder, has been reported as an adverse event in the clinical trial of the approved vaccine in older adults [9]. Although GBS has not been reported in clinical trials in pregnant women, we identify it as a safety event of interest.

In the MATISSE trial, there was no evidence of a difference in the percentage of preterm births between the vaccine and placebo groups overall. However, analyses stratified by income level revealed a statistically significant difference in upper-middle-income countries [10]. Of note, another trial of a candidate maternal RSV vaccine of a similar type stopped early due to a higher risk of preterm birth in the vaccine group [11]. As the first vaccine to be approved for maternal indication and given this potential uncertainty about preterm birth, the regulatory agencies each approved the vaccine to be used at different gestational ages [12]. The EMA decided that the vaccine could be administered beginning at 24 weeks of gestation. The FDA approved vaccination of pregnant women at 32 through 36 weeks of gestation, while the MHRA has recommended that the vaccine may be given in the third trimester of pregnancy (weeks 28 to 36 of gestation) [13].

In light of the importance of addressing gaps in safety data, a protocol for a PASS to assess the safety of this RSV vaccine in pregnant women and their offspring in real-world settings was developed as an additional pharmacovigilance activity outlined in the EU and UK-approved risk management plans and as a post-marketing requirement to the FDA [14,15]. This manuscript provides a generic framework of the PASS protocol to assist researchers in the preparation of their own protocols during the post-marketing phase in the coming years.

## 2. Objectives

The primary objective of a PASS on maternal RSV immunisation study is to estimate the occurrence of adverse maternal, pregnancy and birth outcomes in women who receive the vaccine during pregnancy (and their offspring), compared with a matched comparator group of pregnant women who do not receive the RSV vaccine during pregnancy (and their offspring). The secondary objectives may include assessing the occurrence of these adverse outcomes in immunocompromised or high-risk pregnant women who receive the RSV vaccine compared to those who do not, as these groups were not studied in clinical trials. Due to variations in timing of authorised administration, another objective may be to estimate the occurrence of adverse outcomes by gestational week of vaccination to better understand possible effect modification by week of gestation.

## 3. Methods

### 3.1. Study Design

We recommend an observational retrospective comparative cohort study of pregnant women who receive the RSV vaccine compared to an unexposed pregnant comparator group (or an active comparator group when there is one). To assess whether the RSV vaccine increases the risk of any adverse outcome in pregnancy or offspring, we propose designing the study to emulate a hypothetical pragmatic randomised trial (Table 1) [16]. This trial would assess the safety of administering one dose of the RSV vaccine between 24 and 36 weeks of gestation in pregnant women compared to pregnant women who do not receive an RSV vaccine (Figure 1A). As we cannot truly randomise participants, we need to emulate this feature by carefully addressing confounding biases through various techniques such as matching and careful assessment of inclusion criteria. These techniques also have the potential to address both selection bias and immortal time bias. Time zero, designated as day 0, marks the point at which the eligibility criteria are fulfilled, and the RSV vaccine is administered to the pregnant woman.

At Time zero, exposed participants should be matched to unexposed pregnant participants eligible to receive the vaccine (1:N ratio) based on (Figure 1B):Gestational age (same week of gestation) to account for timing of vaccination in pregnancy and the risk of preterm birthCalendar time (same week) because of RSV seasonality and probability of exposureMaternal age (year of birth) because of the association with adverse pregnancy outcomesImmunocompromised status because of the association with adverse maternal outcomesHigh-risk pregnancy because of the association with adverse pregnancy outcomes

Additional characteristics may be added that would confound the association.

A primary analytical concern is the gestational age at Time zero, which directly impacts the risk of preterm birth. Therefore, it is imperative to ensure a robust match on gestational age, and in fact, we recommend emulating a series of separate target trials for different gestational ages at vaccination, i.e., 24, 25, 26, …, or 36 weeks to study effect modification [17]. Follow-up of the exposed and unexposed (or active comparator) pregnant women should start on day 1, and end at the earliest of N months after birth depending on the outcome being measured, maternal death, disenrollment or migration, end of data availability in the data source, treatment crossover, or occurrence of a given outcome. Treatment crossover, or protocol violation, occurs when previously unexposed pregnant women in the matched comparator cohort receive vaccination with an RSV vaccine during pregnancy. To uphold statistical power, unexposed pregnant women within the matched comparator cohort ought to be censored upon treatment violation, while exposed pregnant women should remain uncensored (see Section 3.10). When unvaccinated pregnant women receive an RSV vaccine, they become eligible for inclusion in the vaccinated cohort (Figure 1C). Follow-up of offspring to assess birth outcomes will depend on the risk window for measuring the outcome and will end at the earliest of N months of age, neonatal death, disenrollment or migration, end of data availability in the data source, or occurrence of a given outcome.

### 3.2. Study Setting

To have sufficient power, we recommend using real-world secondary data derived from electronic health records or administrative claims data that can be linked on an individual level to birth registries and outcomes in offspring (mother-child linkage). The specific study setting will be defined by the availability and nature of the data source(s) utilised and will depend on the country or region covered by the data source. To assess whether data sources are fit for purpose, we stress the importance of ascertainment the accuracy of estimating the start and end of pregnancy, the ability to detect ongoing pregnancies, the capacity to identify exposure to RSV vaccines including timing of vaccine dose, the feasibility of linking mother and child, the capability to measure outcomes and confounders of interest, and the lag time for data availability for each study component. We recommend the use of quality indicators and external benchmarks to assess whether a data source is fit for purpose [18,19]. Key aspects of this evaluation include:Population coverage: The data must cover a representative sample of the population under study. This includes demographics such as age, sex, ethnicity, and geographic location.Data granularity: The data must be granular enough to capture individual-level information on healthcare encounters, including diagnoses, procedures, prescriptions, and outcomes.Longitudinal data: Longitudinal data, tracking individuals over time, is essential for understanding disease progression, treatment patterns, and outcomes.Data quality: High-quality data is crucial for reliable analysis. This involves accurate information recording, standardised coding systems, and measures to address missing or erroneous data.Data linkage: The ability to link data across different sources (e.g., primary care, hospitals, pharmacies, birth registers) enables a comprehensive view of healthcare utilisation and outcomes.Privacy and ethical considerations: Data sources must adhere to strict privacy regulations and ethical guidelines to protect patient confidentiality and ensure informed consent.Timeliness: Timely updates ensure that researchers can access the most current information, allowing for real-time or near-real-time analyses.

### 3.3. Study Period

The first maternal vaccine was launched on 23 August 2023 [4]. In a hypothetical scenario focused on a study of its safety, we propose commencing data collection retrospectively from 24 August 2022. This means that gathering data starting from the day following approval, allowing for a one-year look-back period. This period is essential for capturing prior medical history, including diseases, comorbidities, and medication use, which are critical factors to consider in the study’s analysis and interpretation. Data collection would continue until the last data availability for each data access provider based on the calendar date of the last data extraction. Given the anticipated need for a significant number of participants, we expect a follow-up period of at least 5 years, with the latest data extraction planned for 2029. However, we recommend regular progress reports, with reporting triggered by the number of exposures.

### 3.4. Study Population

The study population comprises all pregnant women who receive an RSV vaccine during pregnancy from the earliest indicated timing of gestation for which the vaccine is licensed and a matched comparator group of pregnant women who did not receive the RSV vaccine or received another RSV vaccine. Identification of pregnancies may vary across data sources which is very important since most data sources that link to birth registries only detect pregnancies when they end. Validated data source-specific algorithms or the ConcePTION pregnancy algorithm could be employed to determine the start and end dates of ongoing pregnancies. The ConcePTION algorithm was created to identify ongoing pregnancies using diagnosis and procedure data related to pregnancy [20]. The identification of ongoing pregnancies is important to avoid selection bias towards pregnancies that end prematurely. In instances where pregnancy data are only available when the pregnancy has ended, a minimum of 10 months of administrative follow-up from the last menstrual period (LMP) in the data source, unless they died, is recommended for all pregnancy and birth outcomes. This period covers the gestational period plus an additional month to identify offspring outcomes at birth [17]. It is important to anticipate that women may have more than one pregnancy, and those subsequent pregnancies could be included to increase power.

### 3.5. Recommended In- and Exclusion Criteria

Participants are eligible if they:have an identified start of pregnancy date.are enrolled in the healthcare system for a least 12 months prior to Time zero (date of vaccination).have at least one day of follow-up after Time zero for maternal outcomes and at least 10 months of follow-up after the LMP for birth and pregnancy outcomes.

Participants are excluded if they:are exposed to an RSV vaccine prior to the current pregnancy or before or after the gestational week for which the vaccine is licensed (vaccination in a prior pregnancy is permitted).cannot be linked in the data source to their child (mother-child linkage).

### 3.6. Exposure

The exposure of interest is RSV vaccination during pregnancy. The vaccine data could be obtained from pharmacy dispensing records, general practice records, immunisation registers, vaccination records, medical records or other databanks. We will consider a single administration of one dose between 24–36 weeks of gestation. An woman is considered exposed from the date of vaccination until the end of pregnancy for all pregnancy and birth outcomes.

### 3.7. Outcomes

The proposed key outcomes of interest, along with their respective outcome-specific exposure windows, are described in Table 2 and encompass preterm birth, time between vaccination and birth, stillbirth, hypertensive disorders of pregnancy, Guillain-Barré Syndrome (GBS), low birth weight (LBW), and small for gestational age (SGA). The outcome time between vaccination and birth holds more statistical power than preterm birth, as preterm birth is a categorical outcome while the time between vaccination and birth is continuous [21,22]. Distinguishing between outcomes measured during pregnancy and those assessed at or after birth is crucial, as it affects statistical analysis and interpretation.

Additional outcomes for maternal immunisation may be added based on the risk profile of the vaccine [30]. The clinical definitions for outcomes of interest should be based on either the Global Alignment on Immunization Safety Assessment (GAIA), Brighton Collaboration, or the official World Health Organization (WHO) definitions [31,32]. While these definitions can be adjusted, utilising these official, globally recognised definitions will enhance harmonisation and study comparability. The outcomes should be ascertained using algorithms based on codes for diagnoses, procedures, medical products and information from each specific databank. We recommend working with each data source to understand how information can be retrieved and how outcomes and other variables are coded.

### 3.8. Covariates

Crucial covariates of interest include matching variables: gestational age, calendar time, maternal age, immunocompromised status and high-risk pregnancy, for which definitions are provided in Table 3. Similar to outcomes, we recommend covariates to be identified using diagnoses, procedures, medical products and information from each specific databank. Additionally, variables associated with the outcomes and exposure of interest should be included [33,34,35,36,37,38]. An example is presented in Appendix A. Maternal vaccinations such as influenza, SARS-CoV-2, Tetanus, Diphtheria and Pertussis (Tdap) given prior to RSV vaccination should also be recorded, as well as RSV monoclonal antibodies and RSV vaccination of the offspring.

### 3.9. Study Size

All women meeting eligibility criteria during the study observation period should be included in the study. It is currently unknown how many women will receive RSV vaccination during pregnancy, as the National Immunisation Technical Advisory Groups need to determine whether this will be included in routine maternal immunisation programs in the country or region where the study is conducted. Minimal detectable risks can be calculated based on the assumption that every vaccinated pregnant individual could be compared with N unvaccinated pregnant individual(s) (1:N ratio). The desired alpha level is 5%, and risk ratio (RR) values of 1.5, 2.0, 2.5, and 4.0 may be considered (Appendix A). For pregnancy and birth outcomes, the required number of exposed women to achieve a power of 80% under different RR values may assume prevalence rates in the unexposed group ranging from 0.2% to 6.0%. To detect an RR of 1.5 with 95% confidence interval (CI) for preterm birth, approximately 1200 pregnancies ending in live birth would need to be included in a data source. For rare outcomes during pregnancy such as GBS, we calculated the power obtainable assuming a maximum sample size of 30,000 exposed pregnant women, with an incidence rate of 1/100,000 person-years in the comparator group [39].

### 3.10. Data Analysis

Detailed methodology for summary and statistical analyses should be documented in a statistical analysis plan and be drafted prior to any analyses. Any major modifications to the prespecified analyses should be clearly reported. Changes should be annotated and amended when they are major.

#### 3.10.1. Baseline Characteristics

Baseline characteristics for both maternal RSV-exposed and pregnant comparator cohorts should be reported as means, standard deviations, medians and other quartiles for continuous variables, and counts and proportions for categorical variables. Any missing baseline characteristics and the duration of the look-back period should be explicitly described. To assess the comparability of matched cohorts, a standardised difference between the index and comparator cohort can be computed for each baseline characteristic. In cases where categorical variables have more than two levels, an overall standardised difference should be calculated.

#### 3.10.2. Measures of Effect

Primary analyses should estimate counts, proportions, and risk or rate ratios with 95% CI for events occurring and measurable during pregnancy, and counts, proportions, and prevalence ratios with 95% CI for outcomes measurable at birth. Comparison of outcomes by exposure status should be conducted using multivariable generalised linear models. These analyses should control for potential confounding by matching. Balance of covariates should be analysed using standardised mean differences (SMDs), some characteristics and risk factors for the exposure or outcomes may remain unbalanced, which should be analysed using plots of SMDs. To address this, propensity score methods can be employed and propensity scores can be adjusted or additionally matched on if necessary [40]. To obtain more power for the analysis of preterm birth, we recommend that time from vaccination to birth should be analysed separately [21,22].

The primary analyses may face-selective censoring due to treatment crossover in the comparator cohort, necessitating the use of inverse probability censoring weights. To maintain statistical power, unexposed pregnant women in the matched comparator cohort should be censored upon treatment violation, while exposed pregnant women should not be censored. Weights can be used to correct for the bias introduced by such censoring, ensuring more accurate and reliable results. Additionally, a sensitivity analysis can be conducted to explore the effect of censoring the matched pair if treatment crossover happens in the previously unexposed pregnant woman. Another sensitivity analysis may be carried out to explore the effect of exclusion of medically indicated iatrogenic births.

For subgroup analyses, secondary analyses can be conducted among immunocompromised or high-risk pregnancies and stratified by weeks of gestation at the time of vaccination.

#### 3.10.3. Meta-Analysis

In case multiple data sources are utilised, we recommend employing a common analytics tool, such as R studio, to conduct the analyses separately within each data source consistently. Subsequently, utilising the main estimates obtained from each data source, appropriate random-effects meta-analytic methods should be applied to derive a combined effect estimate [41,42]. To assess potential heterogeneity across data sources, it is advisable to visually inspect and check using forest plots, displaying the study site estimates and a pooled estimate with 95% CIs. In situations where there are zero events, we suggest pooling incidence rates, as they can be zero and still yield valid standard errors.

### 3.11. Quality Control

We want to highlight that it is important to conduct quality control procedures to ensure the existence and accuracy of records. Quality control can be conducted using the INSIGHT level 1–3 data quality checks [43]. Briefly, level 1 verifies data completeness, level 2 ensures data consistency and level 3 checks for study variables and whether data are fit for purpose.

### 3.12. Protection of Human Subjects

Ethical considerations are paramount in any research involving human subjects. In this generic protocol for a retrospective database study, we describe data that exist in a deidentified/pseudonymised structured format without including personal patient information, thus eliminating the need for informed consent from participants. However, in the event of conducting a new study, it is essential to adhere to ethical guidelines and obtain approval and consent to safeguard the rights and well-being of participants if needed. There must be prospective approval from relevant Institutional Review Boards or Ethics Committees for the study protocol and amendments. Furthermore, we highly recommend conducting the study according to the guidelines for Good Pharmacoepidemiology Practice and the European Network of Centres for Pharmacoepidemiology and Pharmacovigilance (ENCePP) guide on Methodological Standards in Pharmacoepidemiology [44,45]. Additionally, the study protocol should be registered before data collection begins and adhere to transparency and scientific independence principles outlined by the ENCePP Code of Conduct [46].

## 4. Discussion

This generic protocol has been developed to assess the safety of RSV vaccines in pregnant women and their offspring in real-world settings utilising secondary sources of data. It presents the emulation of a target trial, assessing the safety of administering a single dose of an RSV vaccine to pregnant women between 24–36 weeks of gestation compared to pregnant women who do not receive RSV vaccination during pregnancy [8]. The emulation of target trial conditions will help to reduce bias due to censoring, immortal time, competing events and confounding, improve the interpretability of estimands and highlight any remaining challenges with the data [17]. However, despite efforts to minimise bias and improve interpretability, limitations persist in target trial emulation using observational data. These may include challenges in accurately capturing all relevant confounders. Data sources that routinely collect data often provide comprehensive information on treatments and outcomes but may lack sufficient detail on clinical factors that require adjustment [47]. Fitness for purpose of data can be assessed in feasibility studies [18,19].

The study will use code lists or observations in birth registers to identify the covariates and outcomes of interest. An important consideration of using code lists is the potential for outcome misclassification, which can introduce bias and impact the interpretation of findings. The code lists could be sourced from Vaccine Monitoring Collaboration for Europe (VAC4EU) (https://vac4eu.org/), as these have undergone thorough review by medical professionals. Only codes labelled as “narrow” are included to mitigate the false-positive rate. For preterm birth, validation may be needed since preterm birth can either be spontaneous or medically indicated (iatrogenic), whereas our focus is only on spontaneous preterm birth. Codes and outcome definitions may be scrutinised in Safety Platform for Emergency Vaccines companion guides. Validation of outcomes, using Brighton or WHO definitions, may be conducted when resources are available.

In addition to outcome misclassification, it is also crucial to address exposure misclassification. Since RSV vaccination has only recently received approval for maternal immunisation, and many countries’ National Immunisation Technical Advisory Groups have yet to decide on its inclusion in routine immunisation schedules, it can be challenging to determine whether current data sources accurately capture RSV vaccination. This challenge is reflected by the current situation in Europe, where other maternal immunisation practices vary widely across countries, and some prefer to provide monoclonal antibodies or vaccines to neonates [2,48] Therefore, before commencing data collection, it is imperative to liaise with relevant public health organisations to understand if and how RSV vaccination will be integrated into routine immunisation programs for maternal immunisation and in which settings it is provided. Subsequently, it is essential to verify if the data source adequately captures this information and compare it with national statistics to ensure comprehensive coverage and prevent exposure misclassification in the comparator group.

It is important to note that a non-exposed comparator group may be chosen because presently there is only one approved maternal vaccine for preventing RSV infection in offspring. In the event that other RSV vaccines receive approval, it is crucial to consider their inclusion in the study design. This necessitates either incorporating all different RSV vaccines in the exposed group or excluding women specifically vaccinated with other RSV vaccines from the unexposed group.

Because of safety concerns, the primary outcome measure under examination is preterm birth, considering the varying recommendations by EMA, FDA and MHRA regarding the optimal gestational age for vaccination [5,6,7,12,13]. Notably, concerns over preterm births have led to the suspension of a phase 3 trial for another candidate RSV prefusion subunit vaccine [10]. However, a significant challenge in analysing preterm birth in observational studies lies in its direct relationship with gestational age at the time of vaccination. For instance, administering the vaccine at 35 weeks of gestation cannot result in an preterm birth of less than 35 weeks. Therefore, we emphasise the critical importance of matching participants based on gestational age at the time of vaccination. In fact, participants will be enrolled in a sequence of weekly trials based on their gestational age at the time of vaccination. Additionally, we have added the outcome time between vaccination and birth. This outcome holds more statistical power than preterm birth, as preterm birth is a categorical outcome while time between vaccination and birth is continuous. Also, this approach allows us to examine the median time between vaccination and preterm births, which is crucial for identifying specific underlying mechanisms. For instance, it can reveal whether preterm infants are born days, weeks, or months after vaccination. Moreover, this outcome is independent of the accuracy of gestational age assessment [21,22]. Other outcomes can be chosen, and we refer to the GAIA initiative, which defined more than 20 outcomes relevant to maternal immunisation [31].

## 5. Conclusions

This generic protocol is designed to offer guidance for researchers and health authorities in evaluating the safety of RSV vaccination during pregnancy. A target trial emulation approach of observational data can be selected to minimise bias and improve interpretability. We envision this protocol serving as a valuable tool, fostering alignment of research efforts and aiding in the monitoring of the safety of RSV vaccination in pregnant women and their offspring.

## Figures and Tables

**Figure 1 vaccines-13-00272-f001:**
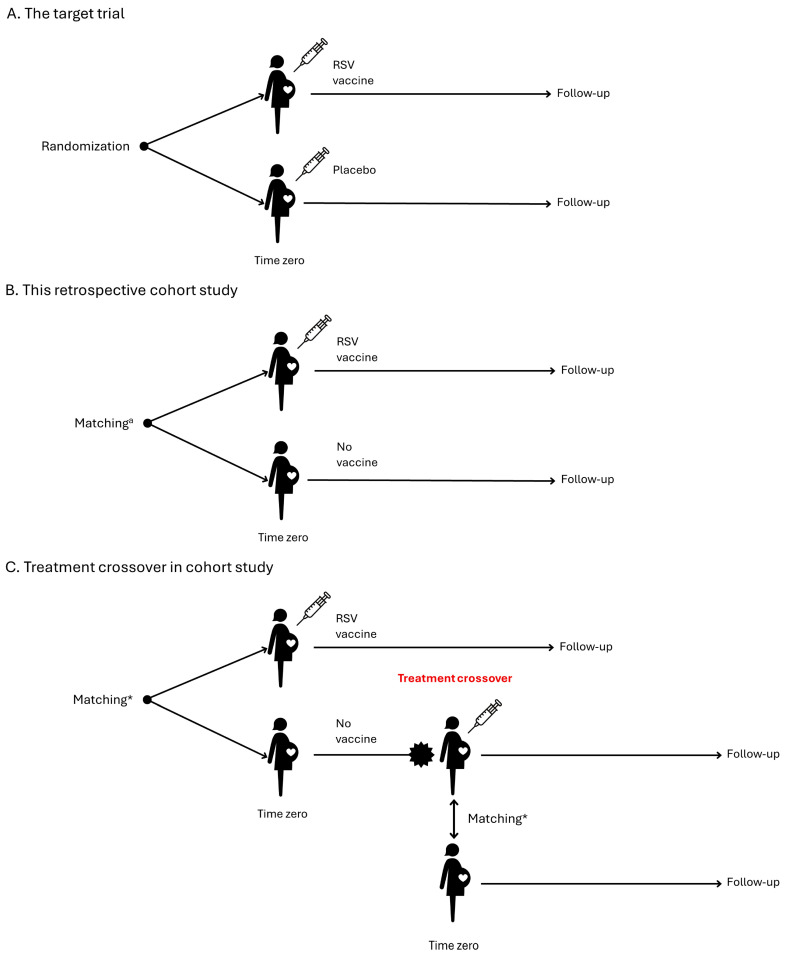
Study design. * Matching should be on gestational age (same week of gestation), calendar time (same week) and maternal age. Additional characteristics may be added.

**Table 1 vaccines-13-00272-t001:** A summary of the protocol of the MATISSE trial and emulation of this target trial to estimate the effect of RSV vaccination during pregnancy.

Protocol Component	Target Trial	Emulation
Eligibility criteria	Healthy women, 49 years of age or younger, at 24 through 36 weeks’ gestation on the day of planned injection, with an uncomplicated, singleton pregnancy and no known increased risk of pregnancy complications.	We maintain the 24 through 36 weeks gestation requirement but do not apply strict age restrictions or restrictions regarding pregnancy complications. Instead, we stratify the analysis based on high-risk pregnancies.
Treatment strategies	(1) A single intramuscular injection of 120 μg of RSVpreF vaccine (60 μg each of RSV A and RSV B antigens) (2) Placebo	(1) RSV vaccination (2) No RSV vaccination
Assignment procedures	Women were randomly assigned in a 1:1 ratio to receive either the RSV vaccine or a placebo, and they were not aware of the strategy to which they had been assigned.	Exposed women will be matched to unexposed women (1:N ratio) based on: Gestational age (same week of gestation) Calendar time (same week) Maternal age (year of birth) Immunocompromised status High-risk pregnancy
Follow-up period	For each maternal participant, including her fetus, data on adverse events were collected from the time of informed consent to 1 month after injection, and data on serious adverse events were collected from the time of informed consent through 6 months after delivery. Safety end points in the infant participants included adverse events from birth to 1 month of age. Additional safety end points were serious adverse events and newly diagnosed chronic medical conditions from birth through 12 to 24 months of age. For vaccine efficacy outcomes in infants started at 72 h after birth and continued through 12 to 24 months of age.	Follow-up of the exposed and unexposed (or active comparator) pregnant women should start on the day after eligibility criteria are fulfilled and end at the earliest of N months after birth depending on the outcome being measured, maternal death, disenrollment or migration, end of data availability in the data source, treatment crossover, or occurrence of a given outcome. Follow-up of offspring to assess birth outcomes will depend on the risk window for measuring the outcome and will end at the earliest of N months of age, neonatal death, disenrollment or migration, end of data availability in the data source, or occurrence of a given outcome.
Outcome	Safety: The primary safety end points were reactogenicity and adverse events in the maternal participants and adverse events and newly diagnosed chronic medical conditions in the infants. Efficacy: The two primary efficacy end points were medically attended severe RSV-associated lower respiratory tract illness and medically attended RSV-associated lower respiratory tract illness.	For post-authorisation safety study purposes, the focus is solely on safety, encompassing preterm birth, stillbirth, hypertensive disorders of pregnancy, low birth weight (LBW), small for gestational age (SGA), and Guillain-Barré Syndrome (GBS).
Causal Contrasts of Interest	Per protocol analysis	Observational analog of per-protocol effect
Analysis	Per protocol analysis	Same per-protocol analysis, except for restriction to pregnancies without loss-to-follow up

**Table 2 vaccines-13-00272-t002:** Outcomes of interest.

Outcomes	Clinical Definition	Pregnancies Among Which the Outcome Will be Ascertained	Exposure Risk Window for Outcome
**Pregnancy outcomes**			
Preterm birth [23]	Preterm birth is defined by the WHO as babies born alive before 37 weeks of pregnancy are completed. It is further subcategorised based on gestational age: Extremely preterm (less than 28 weeks) Very preterm (28 to less than 32 weeks) Moderate to late preterm (32 to 37 weeks)	Pregnancies ending with live births	From time of vaccination to before week 37 of gestation (36 weeks and 6/7 days)
Time between vaccination and birth	Time between vaccination and birth in days.	Pregnancies ending with live and non-live births	From time of vaccination to time of birth
Stillbirth [24]	Stillbirth is defined by the WHO as the death of a foetus that has reached a birth weight of 500 g, or if birth weight is unavailable, gestational age of 22 weeks or crown-to-heel length of 25 cm. It is further subcategorised based on gestational age: Late foetal deaths (greater than 1000 g or after 28 weeks) Early foetal deaths (500–1000 g or 22–28 weeks)	Pregnancies ending with live and non-live births	From time of vaccination to end of pregnancy
**Maternal outcomes**			
Hypertensives disorders of pregnancy [25,26]	Hypertensives disorders of pregnancy include: Gestational hypertension (defined as systolic blood pressure greater than or equal to 140 mm/Hg and/or diastolic blood pressure greater than or equal to 90 mm/Hg arising de novo at ≥20 weeks’ gestation in the absence of proteinuria or other findings suggestive of pre-eclampsia) Preeclampsia de novo (defined as gestational hypertension accompanied by one or more of the following new-onset conditions at ≥20 weeks’ gestation: (a) Proteinuria (b) Other maternal end-organ dysfunction, including neurological complications, pulmonary oedema, haematological complications, renal insufficiency, or impaired liver function (c) Uteroplacental dysfunction Preeclampsia superimposed on chronic hypertension (among persons with chronic hypertension, development of new proteinuria, another maternal organ dysfunction(s), or evidence of uteroplacental dysfunction HELLP (Haemolysis, Elevated Liver Enzyme, Low Platelet) is a serious manifestation of pre-eclampsia and should be assessed if available.	Pregnancies ending with live and non-live births	From time of vaccination to end of pregnancy
Guillain-Barré Syndrome (GBS) [27]	GBS is defined as an neurological condition in which a person’s immune system attacks the peripheral nerves. It is a rare condition and the cause of it is not fully understood. Most cases follow an infection with a virus or bacteria, e.g., Campylobacter jejuni, cytomegalovirus, Epstein-Barr virus and the Zika virus. In rare instances, vaccinations may increase the risk of people getting GBS.	Pregnancies ending with live and non-live births	From time of vaccination until 42 days following vaccination
**Birth outcomes**			
Low birth weight (LBW) [28]	Low birth weight is defined by the WHO as a weight at birth of less than 2500 g, regardless of gestational age.	Pregnancies ending with live births	From birth until one month after birth
Small for gestational age (SGA) [29]	Small for gestational age is defined by the WHO as a birth weight below the 10th percentile for gestational age and sex. SGA can be caused by placental dysfunction, referred to as foetal growth restriction (FGR), or it can be due to a constitutionally small foetus without any pathological causes. Other possible causes include congenital malformations or infections.	Pregnancies ending with live births	From birth until one month after birth

**Table 3 vaccines-13-00272-t003:** Covariates of interest.

Covariate	Definition
Gestational age at time of vaccination	Gestational age at time of vaccination is defined as the time between the last menstrual period (LMP) and the date of vaccination, measured in weeks.
Calendar time at time of vaccination	Calendar time at the time of vaccination is defined as the specific calendar date of the vaccination, expressed in weeks (e.g., week 48 of the year 2024 or week 12 of the year 2025).
Maternal age	Maternal age at time of vaccination is defined as the age of the pregnant person at the date of vaccination. This age is determined by subtracting the individual’s date of birth from the date of vaccination.
Immunocompromised status	Immunocompromised status is defined as meeting at least one of the following criteria at the time of vaccination or in the period prior to vaccination: Medical conditions (a) Diagnosed with symptomatic HIV/AIDS (b) Diagnosed with hematologic malignancy (e.g., chronic lymphocytic leukaemia, non-Hodgkin lymphoma, multiple myeloma, acute leukaemia) (c) Diagnosed with solid malignancy (d) Diagnosed with rheumatologic/inflammatory conditions (e.g., Sjogren’s syndrome, SLE, psoriatic arthritis, rheumatic arthritis, arthritis spondylarthritis, polymyalgia rheumatica, demyelination multiple sclerosis, polymyalgia rheumatica, IBD, autoimmune thyroiditis) and have evidence of treatment with chemotherapy or immune modulators (see below) Immunosuppressive treatments (a) Organ transplant recipients or islet transplant recipients taking immunosuppressive therapy (b) CAR-T-cell therapy or hematopoietic stem cell transplant recipients taking immunosuppressive therapy (c) Rheumatologic/inflammatory conditions treated with systemic corticosteroids (d) Active treatment (at time of vaccination) with various immunosuppressive agents (e.g., systemic corticosteroids, alkylating agents, antimetabolites, transplant-related immunosuppressive drugs, cancer chemotherapeutic agents classified as severely immunosuppressive, TNF blockers, and other biologic agents that are immunosuppressive or immunomodulatory). Specific medical procedures (a) Organ transplant recipients or islet transplant recipients (b) CAR-T-cell therapy or hematopoietic stem cell transplant recipients
High-risk pregnancy	High-risk pregnancy is defined as pregnancies in which women or their offspring have a history of any of the following conditions in the current pregnancy (prior to vaccination): Obesity Hypertension Pre-eclampsia/eclampsia Multifetal pregnancy Diabetes Gestational diabetes Congenital anomaly And/or any of the following conditions in previous pregnancies: Gestational diabetes Pre-eclampsia/eclampsia Stillbirth or late miscarriage Small for gestational age or foetal growth restriction Congenital anomaly

## Data Availability

The original contributions presented in this study are included in the article/Appendix A. Further inquiries can be directed to the corresponding author.

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
