# Peer review of "A Post-Authorisation Safety Study of a Respiratory Syncytial Virus Vaccine in Pregnant Women and Their Offspring in a Real-World Setting: Generic Protocol for a Target Trial Emulation"

_vaccines, 2025, doi:10.3390/vaccines13030272_

Round 1
Reviewer 1 Report
Comments and Suggestions for Authors
The safety of RSV vaccination during pregnancy has been studied in clinical trials of almost 4,000 women and been given to many thousands of women in national programs. Monitoring in the USA, where it has been given to over 300,000 women, has shown a good safety profile. In the main clinical trial, in the month after vaccination, there were slightly more premature babies in the vaccine group (2.1%) than the group that didn’t have the vaccine (1.9%). The vaccine has been approved by medicines regulators in the UK, Europe and USA on the basis of protection, quality and safety.MATISSE study became the part of FDA review to support the approval of Abrysvo, FDA among study participants 60 years of age and older, one participant developed GBS 7 days following vaccination.
The planned study is another one regarding the safety of RSV vaccinations for pregnant women and their effectiveness in preventing RSV infection in offspring.The publication takes the form of a proposed research protocol, ( PASS study) which has already started in 2023 and is to be continued for 5 years.
COMMNETS
The goal of PASS protocol is not clearly defined. All comments are in bold in the review.The primary objective of a maternal RSV immunisation study is to estimate the occurrence of adverse maternal, pregnancy and birth outcomes in women who receive the vaccine during pregnancy compared with a matched comparator group of pregnant women who do not receive the RSV vaccine during pregnanc.
The PASS protocol therefore sets many endpoints that have not been included in previous studies. The reviewer therefore considers it justified to conduct such a project. For this reasons the protocol designates multiple endpoints as: encompassing preterm birth, stillbirth, hypertensive disorders of pregnancy, low birth weight (LBW), small for gestational age (SGA), and Guillain-Barré Syndrome (GBS).
The study is to last 5 years. The protocol is very nicely graphically presented but is not described in detail. It should be described what the active comparator is used in this study. Is it a placebo??? What kind of placebo you want to use. Why use the placebo if authors
Why is randomization to be used in the first phase of the study and a in time zero retrospective-observational study planned in the next phases? Authors propose to use as a statistical techique matching to evaluate the effect of vaccination group by comparing the nonvaccine group in an observation retrospective study. Matching will be on gestational age (same week of gestation), calendar time (same week) and maternal age.The authors assume that additional characteristics may be added. Protocol should take into account all proposed characteristics before starting the study..As I understand it at Time zero, participants should be matched to unexposed pregnant participants eligible to receive the vaccine or not to receive vaccine.Reviewer ask to clarification on what will placebo and why it is necessary in this study ? It also requires explanation what the comparator is in this study,Is this a placebo? In tab 2 the authors list a significant number of immunological disorders. Is it planned to include ( vaccine?) pregnant women with these diseases? How the authors want to analyze the immunocompromised status of pregnant women. The protocol does not clearly describe the excluding criteria. According to the protocol the vaccine is to be administered 24 through 36 weeks of gestation, which raises concerns of the reviewer. Many scientific perinatal societies suggest limiting the time of vaccination administration to the 32nd–36th week of gestation.The time of vaccination should be change.
The references list includes 40 current very well-chosen items.
Reviewer 2 Report
Comments and Suggestions for Authors
The authors developed a protocol for target trial emulation, to assess the real-world safety of RSV vaccination in pregnant women. This is an interesting approach, which can be supplementary to routine pharmacovigilance and may have significant advantageous implications for public health. The authors show how emulating target trial conditions can minimise confounding and bias by matching RSV-vaccinated pregnant women with unexposed women based on gestational age, calendar time, maternal age, immune-compromised status, and high-risk pregnancy. Key adverse outcomes are selected, and the model can be adapted to include additional outcomes as per vaccine risk profile and recommendations of the Global Alignment of Immunization safety Assessment. This report is well written, excellently presented and therefore suitable for publication, with only minor comments to be addressed as specified below.
General comment:
In their introduction, the authors point to different vaccination protocols depending on the regulators’ prescription of the appropriate gestational age. Then it would follow that proposed studies should be carried out in or are limited to the geographical area defined by its regulator, e.g. UK for MHRA, EU for EMA etc. Do the authors agree? Can this be clarified in the discussion?
Specific comments to be addressed:
Ln 15 etc: To avoid unnecessary ambiguities throughout the Manuscript, please replace ‘pregnant individuals’ with ‘pregnant women’ and ‘individuals’ with ‘women’ and ‘pregnant individual’ with ‘pregnant woman’ and ‘individual’ with ‘woman’.
Ln 44: Medicines and Healthcare products Regulatory Agency (MHRA)
Ln 223: Please clarify the statement regarding exclusion of women vaccinated during an earlier pregnancy. It is not clear if this group is excluded from the trials or not. This may well become an issue for future epidemiological studies when RSV vaccination of pregnant women becomes more widespread.
Ln 270: NITAGs should not be used as its frequency is only 2.
Ln 386: Abbreviation VAC4EU should not be used as its frequency is only 2.
Ln 391: Abbreviation SPEAC should not be used as its frequency is only 1.
